# β-Glucan reprograms alveolar macrophages via neutrophil/IFNγ axis in a murine model of lung injury

Renaud Prevel[1], Erwan Pernet[1], Kim A Tran[1], Abderrahmane Sadek[2,3], Mina Sadeghi[1], Elizabeth Lapshina[1], Leonardo F Jurado[1], Arnold S Kristof[1], Mohieddine Moumni[3], Jeremie Poschmann[2], Maziar Divangahi[1,4,5,6]*

[1]Department of Medicine, Meakins-Christie Laboratories, Research Institute McGill University Health Centre, McGill University, Montreal, Canada; [2]INSERM, Nantes Université, Center for Research in Transplantation and Translational Immunology (CR2TI), UMR 1064, ITUN, Nantes, France; [3]Biotechnology and Bioresources Valorization Laboratory, Biology Department, Faculty of Sciences, Moulay Ismail University of Meknès, Meknès, Morocco; [4]Department of Microbiology and Immunology, Meakins-Christie Laboratories, Research Institute McGill University Health Centre, McGill University, Montreal, Canada; [5]Department of Pathology, Research Institute McGill University Health Centre, McGill University, Montreal, Canada; [6]McGill International TB Centre, Montreal, Canada

*For correspondence:
maziar.divangahi@mcgill.ca

Competing interest: The authors declare that no competing interests exist.

## eLife Assessment

This **important** study advances our understanding of maladaptive innate immune training. The experimental evidence supporting the conclusions is **convincing** and the expert reviewers strongly endorse the manuscript. The work will be of high interest to both researchers in the trained immunity field and clinician scientists.

**Abstract** Alveolar macrophages (AMs) reside in the lower airways and play a crucial role in lung health and response to sterile inflammation and infections. AMs possess remarkable adaptability to different environmental challenges that can persist through their memory capacity (trained immunity). β-Glucan has been characterized as a potent inducer of central trained immunity by reprogramming haematopoietic stem cells in the bone marrow. In the present study, we show that systemic administration of β-glucan in mice induces peripheral trained immunity by reprogramming AMs in the lungs, in a Dectin1-independent manner. We furthermore demonstrate that AM reprogramming at both the transcriptional and metabolic levels exacerbate lung injury following bacterial (lipopolysaccharide) or viral (polyI:C) challenges via a neutrophil/IFN-γ-dependent manner. These findings identify an additional facet of β-glucan in trained immunity involving AM reprogramming and shed light on the potential detrimental effects of trained immunity.

## Introduction

The lungs represent a significant infective niche for numerous infectious agents. However, severe pulmonary disease is increasingly being associated with immune-mediated responses rather than direct damage of invading microbes (*Xiao et al., 2011*; *Gustine and Jones, 2021*). Among the various triggers of immunopathology, lipopolysaccharide (LPS), a component of the outer membrane

of Gram-negative bacteria, is a potent inducer of systemic inflammation via activation of Toll-like receptor 4 (TLR4) (*Opal, 2010*). Excessive inflammation associated with LPS can cause acute lung injury (ALI), a severe inflammatory condition of the lungs characterized by alveolar damage, edema, and impaired gas exchange leading to organ dysfunction, and ultimately, mortality (*Xu et al., 2024*; *Gill et al., 2015*). Thus, the magnitude of inflammatory responses must be regulated to resolve infections while preventing collateral tissue damage. Understanding the mechanisms underlying the regulation of inflammation in ALI is essential for the development of effective therapeutic interventions. Key players in this regulatory network include various immune cells, cytokines, and signalling pathways, which collectively modulate the intensity and duration of inflammatory responses. Dysregulation of these mechanisms can lead to either persistent inflammation or immunosuppression, both of which contribute to the pathogenesis of sepsis (*Sun et al., 2023*; *Kumar, 2020*; *Zhou and Liao, 2021*).

The constant exposure of the lungs to a non-sterile environment leads to the development of a unique immunity by eliminating inhaled foreign particles while minimizing inflammatory-mediated tissue damage. Resident alveolar macrophages (AMs) serve a crucial regulatory function to maintain the delicate balance between inflammation and protecting lung tissue from damage (*Schneider et al., 2014*). AMs primarily originate from fetal liver monocytes that colonize the airways during development and subsequently maintain their population through local self-renewal. The pulmonary AM pool can also be replenished beyond the neonatal period by bone marrow-derived monocytes following different stimuli (*Misharin et al., 2017*; *Li et al., 2022*). Due to their strategic location, AMs serve as vigilant sentinels, protecting the airways from invading pathogens and pollutants. They play a pivotal role in orchestrating both the initiation and resolution of immune responses within the lung microenvironment (*Guilliams et al., 2013*; *Hussell and Bell, 2014*).

Adaptations in innate immune cells are diverse with substantial plasticity to various insults that can be maintained resulting in enhanced (trained immunity) or reduced (tolerance) inflammatory responses to a second stimuli (*Divangahi et al., 2021*; *Netea et al., 2020*; *Netea et al., 2016*). Trained immunity is mediated via long-term metabolic reprogramming and epigenetic modifications, which can be induced by various stimuli such as the attenuated mycobacteria Bacille Calmette-Guérin (BCG) or β-glucan (a polysaccharidic component of fungi cell wall) (*Divangahi et al., 2021*; *Camilli et al., 2018*). BCG and β-glucan are able to train monocytes and neutrophils with beneficial impact in cancer or infections via the reprogramming of haematopoietic stem cells (HSCs) within the bone marrow (*Moorlag et al., 2020*; *Kaufmann et al., 2018*; *Roquilly et al., 2020*; *Broquet et al., 2024*; *Khan et al., 2025*). However, there are substantial knowledge gaps in our understanding of trained immunity within tissue-resident macrophages and its regulatory role in maintaining tissue homeostasis under stress conditions. For instance, while the initial LPS stimulation of macrophages induces a strong inflammatory responses via TLR-4, restimulation with LPS generates tolerance in these macrophages (*Foster et al., 2007*). Thus, the epigenetic reprogramming of innate immune cells and their subsequent response depends on two signals: the initial training agent and the nature of the second stimuli at an inflammatory site. Here we showed that systemic administration of β-glucan can reprogram AMs in the lungs via neutrophil/type II IFN axis but independent of Dectin-1 signalling. The unique transcriptomic and metabolic profile of AMs render them hyperresponsive to both bacterial and viral stimulation causing dysregulated inflammatory responses with pulmonary damage. The differences in the systemic basal levels of β-glucan in sepsis patients as well as the pulmonary levels of IFNγ can be major factors in determining the hyperresponsiveness in sepsis patients and can be potentially targeted for therapy.

## Results

### β-Glucan-mediated trained immunity aggravates LPS-induced ALI

To investigate whether systemic administration of β-glucan induces trained immunity within the lung, we sought to assess its impact on ALI triggered by LPS treatment. We initially evaluated the consequences of β-glucan-mediated training on ALI by administering LPS 7 days after training (*Figure 1A*). ALI assessment conducted 24 hr post-LPS instillation did not reveal any discernible distinctions attributable to the initial i.p. β-glucan injection. However, ALI was notably exacerbated in mice that had undergone β-glucan training upon subsequent LPS stimulation as demonstrated non-invasively by lung microCT scanners with regard to the increased proportion of poorly or non-aerated lung segment

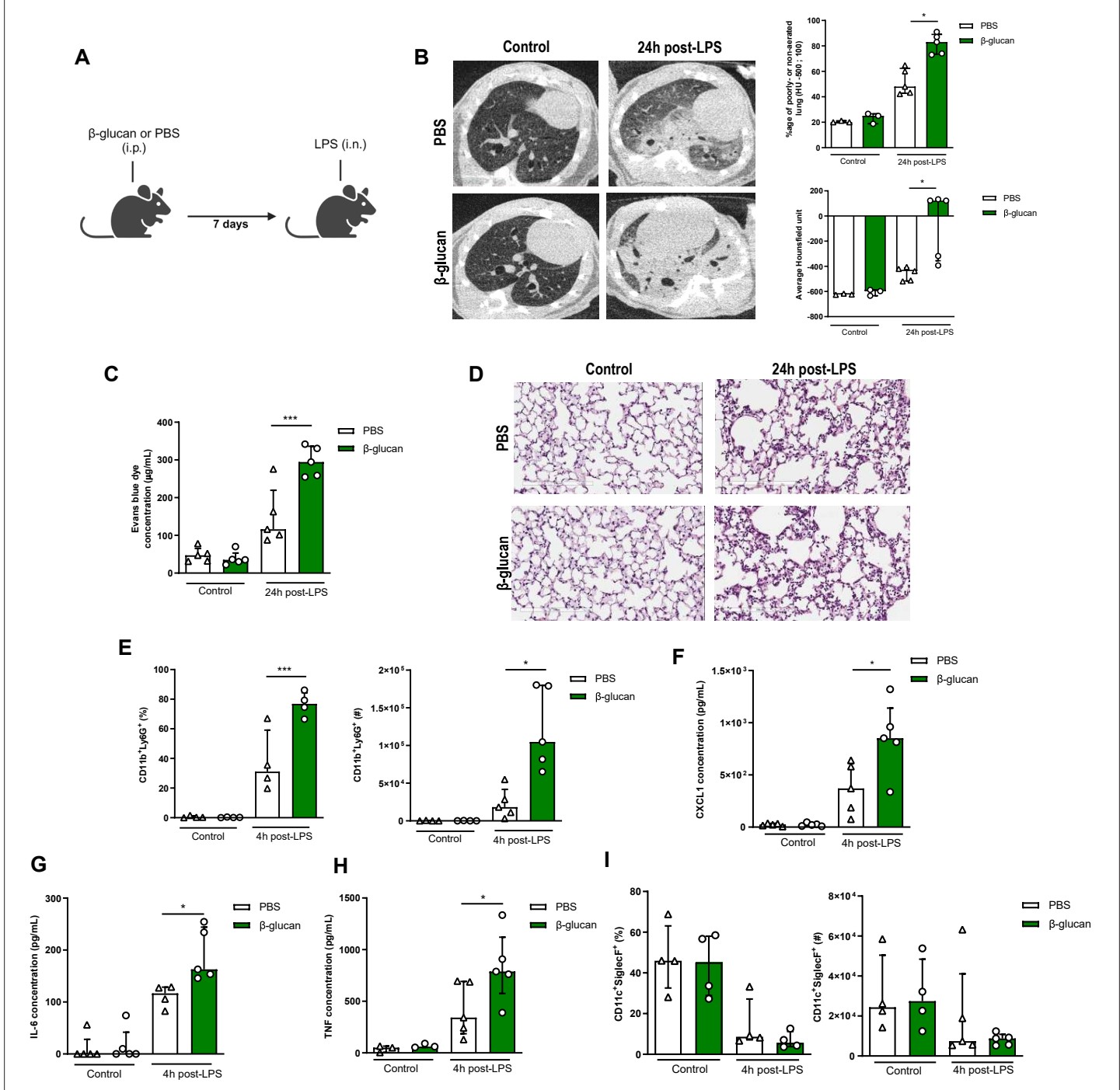

**Figure 1.** β-Glucan-mediated trained immunity increases lipopolysaccharide (LPS)-induced acute lung injury (ALI). (**A**) Schematic of the β-glucan-induced training 7 days before LPS-induced ALI model. Experiments were performed in sex- and age-matched 10- to 12-week-old control (i.p. PBS, white bars) and trained (i.p. β-glucan, green bars) WT mice. (**B**) Lung micro-CT scan, percentage of poorly or non-aerated lung and average lung Hounsfield unit. (**C**) Alveolar capillary membrane permeability assessed by lung Evans blue dye concentration. (**D**) Lung histology after staining with haematoxylin and eosin (Scale bar = 200 μm). (**E**) Quantification of bronchoalveolar lavage (BAL) neutrophils frequency (left) and absolute count (right) (gated on single live cells, CD45.2$^+$, CD11c$^-$, Siglec-F$^-$, CD11b$^+$, Ly6G$^+$). (**F–H**) BAL chemokine and pro-inflammatory cytokines concentrations (left to right) (CXCL1: chemokine C-X-C motif ligand 1, IL-6: interleukin-6, and TNF: tumour necrosis factor). (**I**) Quantification of BAL alveolar macrophages frequency (left) and absolute count (right) (gated on single live cells, CD45.2$^+$, CD11c$^+$, Siglec-F$^+$). Data were analysed using one-way ANOVA followed by Dunn's multiple comparisons test. *p < 0.05, ***p < 0.001. Schematics created using BioRender.com.

The online version of this article includes the following source data and figure supplement(s) for figure 1:

*Figure 1 continued on next page*

*Figure 1 continued*

**Source data 1.** Individual measurements, cytokine concentrations, cell frequencies and cell numbers.

**Figure supplement 1.** Weight loss following lipopolysaccharide (LPS) instillation.

**Figure supplement 1—source data 1.** Animal weights overtime post-LPS.

**Figure supplement 2.** Alveolar macrophage (AM) characterization in β-glucan-treated mice.

**Figure supplement 2—source data 1.** Cell frequencies.

**Figure supplement 3.** Long-term effects of β-glucan-mediated trained immunity on lipopolysaccharide (LPS)-induced acute lung injury (ALI).

**Figure supplement 3—source data 1.** Individual measurements, cytokine concentrations, cell frequencies and cell numbers.

**Figure supplement 4.** Long-term effects of β-glucan-mediated trained immunity on alveolar macrophages (AMs).

**Figure supplement 4—source data 1.** Cytokine concentrations.

**Figure supplement 5.** β-Glucan-mediated trained immunity increases poly(I:C)-induced acute lung injury (ALI).

**Figure supplement 5—source data 1.** Individual measurements, cytokine concentrations, cell frequencies and cell numbers.

and to a significant increase in Hounsfield units (*Figure 1B*). Hounsfield units reflect a coefficient of tissue attenuation and, by extrapolation, tissue composition. Consistent findings were corroborated through two complementary invasive evaluations of ALI. Specifically, the heightened alveolar-capillary permeability following LPS instillation in β-glucan-trained mice was substantiated by elevated lung Evans blue dye concentrations (*Figure 1C*). Histological examination further unveiled a spectrum of notable alterations following LPS instillation, including cellular infiltration, thickened alveolar walls, and the formation of hyaline membranes, observed in β-glucan-trained mice and not PBS controls (*Figure 1D*). We additionally monitored weight loss following LPS instillation. ALI-induced inflammation peaks at 24 hr, reflected in the weight loss of the mice which drops nearly 10% within a day. While there were no differences in weight loss between control and β-glucan-treated mice at this timepoint, we observed that β-glucan mice exhibited a delayed recovery period given they did not return to their original weight by day 4, as opposed to control mice (*Figure 1—figure supplement 1*).

The increased cell infiltration 24 hr after LPS instillation in β-glucan-treated mice was due to a significant recruitment of neutrophils, as assessed in the bronchoalveolar lavage (BAL; CD11b$^+$Ly6G$^+$). Strikingly, the frequency and total numbers of neutrophils was doubled in the β-glucan-treated mice, after LPS administration (*Figure 1E*). Concomitant with the increased neutrophil recruitment, CXCL1 was significantly increased in β-glucan-treated mice when compared to control mice after LPS treatment (*Figure 1F*; *Wengner et al., 2008*). Amplified ALI in β-glucan-trained mice was also associated with increased proinflammatory cytokines (IL-6 and TNF) (*Figure 1G, H*). As AMs are major producers of CXCL1, IL6, and TNF, we next assessed whether β-glucan increased the proportion or number of AMs and found no significant differences between naive and β-glucan-trained mice (*Figure 1I*). There were also no changes in the frequency of CD11b$^+$ AMs at baseline and post-LPS instillation between β-glucan-treated and control mice (*Figure 1—figure supplement 2A*). This suggests that β-glucan does not induce the expansion of monocyte-derived AMs. To further characterize AM phenotype following β-glucan, we assessed several AM-associated cell surface markers and found no differences in expression of MHCII, F4/80, CD169, and CD69 (*Figure 1—figure supplement 2B*). Interestingly, β-glucan enhanced the frequency of CD80$^+$ AMs, a costimulatory molecule associated with activation and pro-inflammatory responses (*Figure 1—figure supplement 2B*; *Burastero et al., 1999*). Thus, the increased cytokine production was not due to an increased number of AMs, but rather intrinsic functional changes.

Next, we examined whether the effects of β-glucan-induced immune training persisted in ALI severity. To do so, we administered β-glucan or PBS (control) in mice and after 28 days challenged them with LPS (*Figure 1—figure supplement 3A*). Similar to the 7-day timeframe of β-glucan training, there was increased endothelial permeability of the lungs, and increased immunopathology (*Figure 1—figure supplement 3B, C*) in 28 days β-glucan-trained mice challenged with LPS. This increased ALI in β-glucan-treated mice after 28 days was associated with a higher concentration of CXCL1, IL-6, and TNF and increased neutrophil infiltration in BAL (*Figure 1—figure supplement 3D, E*) following LPS instillation. The frequency and number of AMs were not changed between groups (*Figure 1—figure supplement 3F*). Increased production of cytokines was also observed upon ex vivo stimulation of AMs isolated from treated mice 28 day post-β-glucan (*Figure 1—figure supplement 4*). Thus, the

intrinsic functional changes in AMs suggest that the exacerbation of ALI induced by β-glucan triggered a long-term reprogramming of immune cells rather than an additive effect of lingering inflammation from β-glucan injection. To assess whether this response was specific to bacterial LPS or viral agonists can cause similar responses, we next challenged β-glucan-trained mice with a TLR-3 agonist (poly(I:C)) (*Figure 1—figure supplement 5A*). Similar to the LPS-model, β-glucan-treated mice had heightened poly(I:C)-induced ALI shown via increased alveolar-capillary permeability, tissue damage (*Figure 1—figure supplement 5B, C*). Pro-inflammatory cytokines CXCL1 and IL-6 were also increased while TNF remains unchanged following challenge suggesting differing cytokine profiles between LPS and poly(I:C)-induced ALI (*Figure 1—figure supplement 5D*). However, β-glucan-treated mice similarly had increased neutrophil infiltration following poly(I:C) instillation (*Figure 1—figure supplement 5E*). The number of AMs was also unchanged among groups (*Figure 1—figure supplement 5F*). Collectively, these results suggest that systemic β-glucan can maintain a long-term reprogramming in AMs promoting ALI.

## β-Glucan augmented ALI is mediated via AM

Although there were no differences in the frequency or absolute number of AMs between β-glucan-treated and control mice, the early heightened response with increased CXCL1 production and the recruitment of neutrophils indicate that the AMs are engaged in the exacerbation of ALI induced by β-glucan. To further characterize the role of these cells in ALI, we locally depleted AMs using intranasally administered clodronate liposomes 2 days before performing the LPS instillation, which is at its peak of AM depletion (*Figure 2A*). Depletion of AMs significantly reduced the production of cytokines TNF, IL-6, and CXCL1 and the recruitment of neutrophils in BAL, which abolished the β-glucan-induced ALI (*Figure 2B–D*).

To further confirm that tissue-resident AMs are responsible for the increased β-glucan-induced ALI, we used *Csf2rb*$^{-/-}$ mice (*Figure 2E*), which do not naturally develop AMs throughout their lifespan but maintain regular levels of bone marrow-derived macrophages and interstitial macrophages (*Robb et al., 1995*). Similar to the AM depletion, the production of cytokines and neutrophil recruitment in BAL was similar between β-glucan-treated and control *Csf2rb*$^{-/-}$ mice after LPS administration. Importantly, there was no difference in ALI, as similar levels of proteins were measured in the BAL of β-glucan-treated and control *Csf2rb*$^{-/-}$ after the LPS challenge (*Figure 2F–H*).

To complement these loss-of-function experiments with a gain-of-function, we adoptively transferred AMs from adult WT control (PBS) or trained (β-glucan) mice to day 2 *Csf2rb*$^{-/-}$ neonate mice (*Figure 2I*) and after 6 weeks challenged these mice with LPS. Remarkably, mice replenished with AMs from β-glucan-trained mice displayed an increased in production of inflammatory cytokines and the recruitment of neutrophils into the BAL (*Figure 2K, L*). This was associated with increased ALI as they had higher protein levels in their BAL compared to mice who received control AMs (*Figure 2J*). Collectively, our data demonstrate that the exacerbation of LPS-induced ALI in β-glucan-trained mice is mediated through AMs, mainly via functional reprogramming rather than a change in the proportion or number of AMs.

## β-Glucan reprograms AM

Our findings indicate that β-glucan reprograms AMs causing an increased response to LPS and subsequently enhanced ALI. To test this hypothesis, we investigated the transcriptional state of AMs after 7 days training with β-glucan in vivo, which were then stimulated with/without LPS ex vivo LPS. The rationale for using an ex vivo system was to ensure that AMs are uniformly stimulated with LPS (*Figure 3A*).

Principal component analysis of all expressed genes across the four conditions revealed that control and β-glucan-trained AMs had distinct transcriptional profiles (*Figure 3B*). More precisely, the four different conditions explained the largest variance as they spread out across the first principal component (PC), while the variation within replicates was associated to the second and third PC. Next, we investigated the effect of β-glucan-trained AMs by comparing the differential gene expression between control and β-glucan-treated mice (*Figure 3C*). We found 88 differentially expressed genes (32 upregulated and 56 downregulated, with a false discovery rate (FDR) adjusted p-value <0.05 and a log$_2$fold-change cut-off of 1). Despite that only few genes were differentially expressed, gene ontology analysis indicated a strong enrichment of genes involved in innate immune response and defense response to virus (*Figure 3D*). Interestingly, LPS response genes were downregulated in

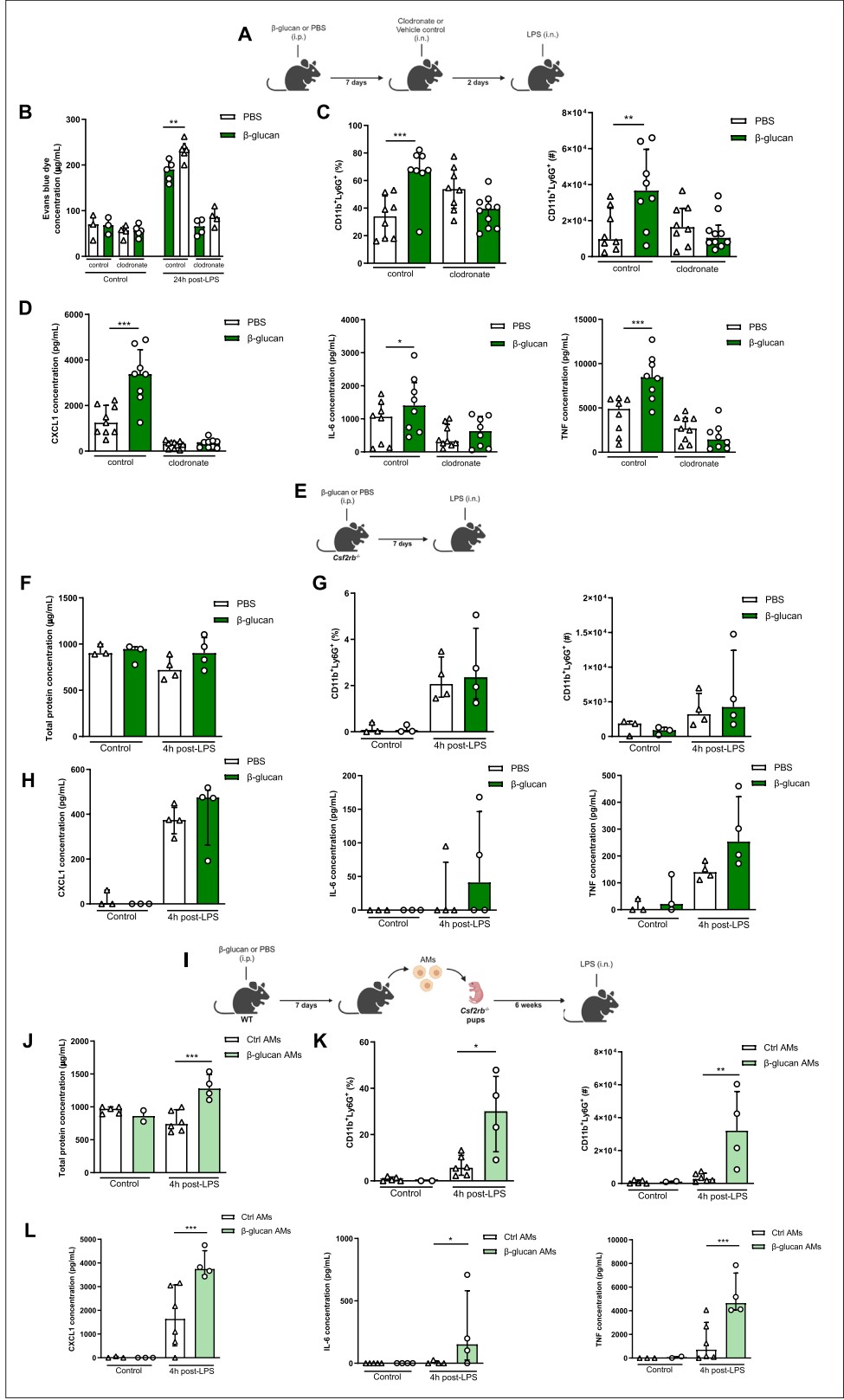

**Figure 2.** Systemic administration of β-glucan enhances acute lung injury (ALI) via alveolar macrophages (AMs). (**A**) Schematic of the clodronate-mediated AMs depletion experiments, performed in sex- and age-matched 10- to 12-week-old control (i.p. PBS, white bars) and trained (i.p. β-glucan, green bars) WT mice. (**B**) Alveolar capillary membrane permeability assessed by lung Evans blue dye concentration. (**C**) Quantification of bronchoalveolar

*Figure 2 continued on next page*

*Figure 2 continued*

lavage (BAL) neutrophils frequency (left) and absolute count (right) (gated on single live cells, CD45.2$^+$, CD11c$^-$, Siglec-F$^-$, CD11b$^+$, Ly6G$^+$). (**D**) BAL chemokine C-X-C motif ligand 1 (CXCL1) and pro-inflammatory cytokines (IL-6: interleukin-6 and TNF: tumour necrosis factor) concentrations. (**E**) Schematic of the β-glucan-induced training and lipopolysaccharide (LPS)-induced ALI model in sex- and age-matched 6-week-old control (i.p. PBS, white bars) and trained (i.p. β-glucan, green bars) *Csf2rb*$^{-/-}$ mice. (**F**) BAL total protein concentration. (**G**) Quantification of BAL neutrophils frequency (left) and absolute count (right) (gated on single live cells, CD45.2$^+$, CD11c$^-$, Siglec-F$^-$, CD11b$^+$, Ly6G$^+$). (**H**) BAL CXCL1, IL-6, and TNF concentrations. (**I**) Schematic of the adoptive transfer of control (i.p. PBS, white bars) or β-glucan-trained (i.p. β-glucan, green bars) AMs collected from adult WT mice to 2 days old *Csf2rb*$^{-/-}$ mice. LPS-induced ALI was performed 6 weeks after adoptive transfer. (**J**) BAL total protein concentration. (**K**) Quantification of BAL neutrophils frequency (left) and absolute count (right) (gated on single live cells, CD45.2$^+$, CD11c, Siglec-F$^-$, CD11b$^+$, Ly6G$^+$). (**L**) BAL CXCL1, IL-6, and TNF concentrations. Data were analysed using one-way ANOVA followed by Dunn's multiple comparisons test. *p < 0.05, **p < 0.01, ***p < 0.001. Schematics created using BioRender.com.

The online version of this article includes the following source data for figure 2:

**Source data 1.** Individual measurements, cytokine concentrations, cell frequencies and cell numbers.

β-glucan-treated mice suggesting that, in response to β-glucan, the LPS response pathway is attenuated or less activated. Gene set enrichment analysis (GSEA) additionally revealed an increase of genes involved in IFNα but a significant decrease of genes involved in TNF signalling (*Figure 3E*).

Next, we explored how ex vivo stimulated AMs responded to LPS using the same differential gene expression analysis. LPS altered the expression of 525 genes (FDR >0.05 log$_2$FC >1.5) with the up regulation of 438 genes including numerous pro-inflammatory genes (*Figure 3F*). As expected, gene ontology analyses revealed that these genes were involved in cellular response to LPS, inhibition of viral genome replication, production of interleukin-1β and cellular response to IL-1 (*Figure 3G*). Additional analysis of GSEA indicated upregulation of genes involved in TNF signalling (*Figure 3H*), indicating that LPS and β-glucan both induce interferon responses but have opposite effects on the TNF response pathway.

To delineate how β-glucan training of AMs may affect the subsequent response to LPS, we then compared the fold change of gene expression in response to LPS stimulation between β-glucan-trained and -untrained AMs. Notably, we observed an upregulation of genes associated with the LPS response (depicted in blue) in the β-glucan-trained group (*Figure 3I*). This heightened response included a significant increase in the expression of IL-6, IL-1, TNF, and other genes (*Figure 3I, J*). This increased response to LPS by β-glucan-trained AMs was confirmed at the protein level as they exhibited increased ex vivo production of CXCL1 and TNF (*Figure 3K*). Such an increased response to LPS was also obtained in AMs after 28 days β-glucan training (*Figure 1—figure supplement 4*). We also observed similar trends in genes associated with the defense response to viruses which is in line with our findings showing an increased ALI to poly(I:C) (*Figure 3—figure supplement 1*). Taken together, β-glucan reprograms AMs to respond robustly to LPS stimulation.

As trained immunity is associated with metabolic rewiring, we next assessed if β-glucan-induced training modified the AM's metabolic state. Genes involved in both oxidative phosphorylation and glycolysis were upregulated during response to LPS in β-glucan-trained AMs compared to untrained AMs but not at the steady state (*Figure 3L*). The Seahorse assay showed while only mitochondrial respiration was increased in β-glucan-trained AMs at the steady state, both mitochondrial respiration and glycolysis were increased during the response to LPS in β-glucan-trained AMs, (*Figure 3M*). Taken together, β-glucan functionally reprograms AMs by rewiring their transcriptomic and metabolic states.

## IFNγ signalling and neutrophils are required for AM reprogramming

Dectin-1 is the receptor that recognizes β-glucan and is involved in mediating the biological effects of β-glucan-induced training in most cell types (*Camilli et al., 2018*). To assess the role of Dectin-1 in β-glucan increased LPS-induced ALI, we used the same experimental setup using Dectin-1-deficient mice (*Figure 4—figure supplement 1A*). Surprisingly, the increased neutrophil infiltration and pro-inflammatory cytokine production in BAL were maintained in β-glucan-treated *Dectin-1*$^{-/-}$ mice and there was no change in the AM numbers (*Figure 4—figure supplement 1B–D*). To determine the effect of *Clec7a* (Dectin-1) on β-glucan-induced AM function, *Dectin-1*$^{-/-}$ mice were treated with

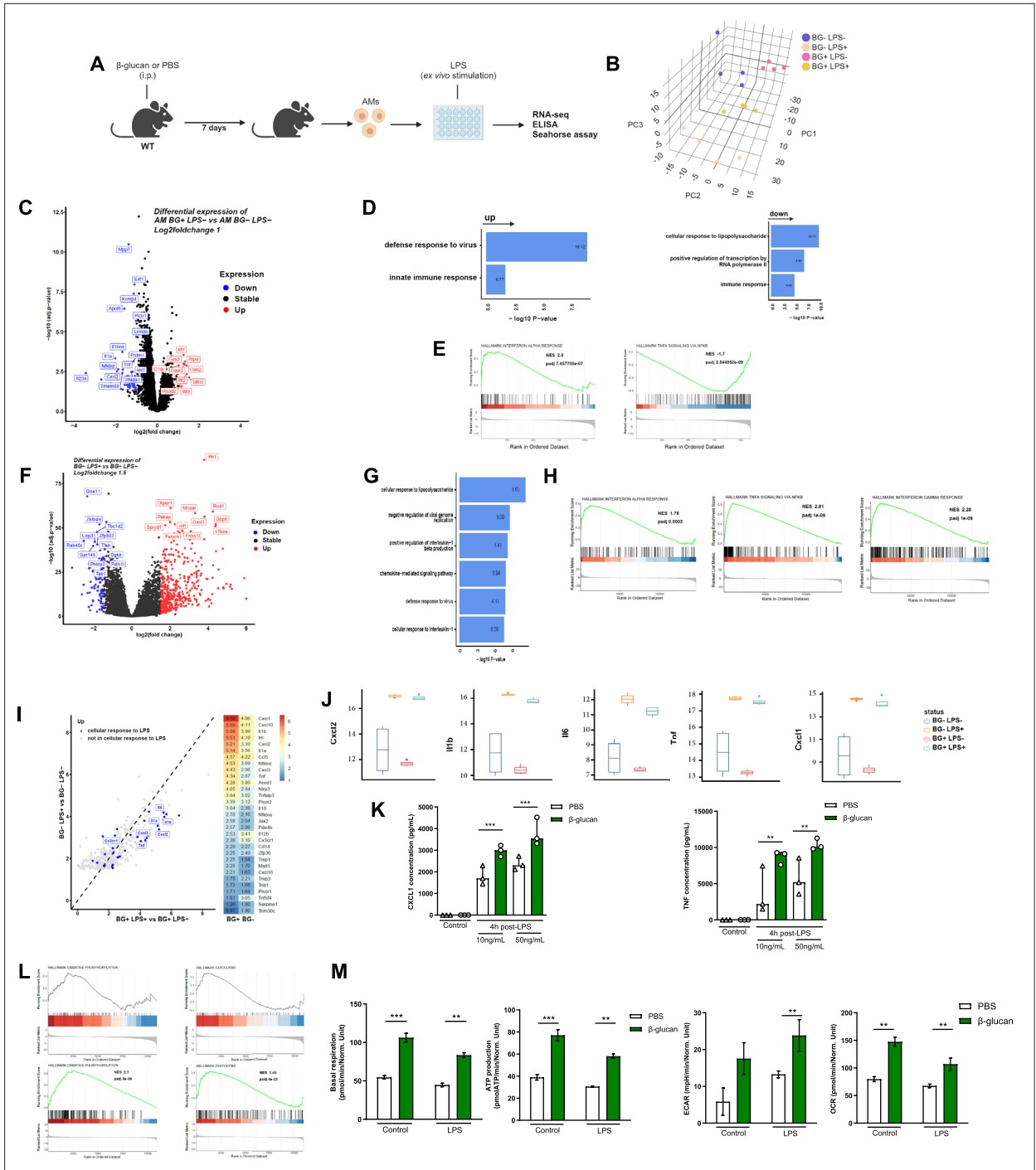

**Figure 3.** β-Glucan reprograms alveolar macrophages (AMs). (**A**) Schematic of control (i.p. PBS) or β-glucan-trained (i.p. β-glucan) AMs collected from adult WT mice ex vivo stimulation with lipopolysaccharide (LPS) (LPS−: unstimulated, LPS+: stimulated in RNA-seq analysis). (**B**) Discovery plot. (**C**) AM differential expression of genes in response to β-glucan training. (**D**) Gene ontology in response to β-glucan training. (**E**) Gene set enrichment analysis (GSEA) in response to β-glucan training. (**F**) AM differential expression of genes in response to LPS stimulation. (**G**) Gene ontology in response to LPS stimulation. (**H**) GSEA in response to LPS stimulation. (**I**) AM gene expression in response to LPS in β-glucan-trained AMs. (**J**) Examples of genes expression in response to LPS in control versus β-glucan-trained AMs. (**K**) Chemokine C-X-C motif ligand 1 (CXCL1) and tumour necrosis factor (TNF) concentrations after ex vivo LPS stimulation. (**L**) GSEA of oxidative phosphorylation (left) and glycolysis (right) pathways according to β-glucan-training in unstimulated (LPS−) and LPS stimulated (LPS+) AMs. (**M**) Evaluation of AM metabolism: basal respiration (upper left), ATP production (upper right), extracellular acidification rate (ECAR, lower left), and oxygen consumption rate (OCR, lower right). Data were analysed using one-way ANOVA followed by Dunn's multiple comparisons test. **p < 0.01, ***p < 0.001. Schematics created using BioRender.com.

*Figure 3 continued on next page*

*Figure 3 continued*

The online version of this article includes the following source data and figure supplement(s) for figure 3:

**Source data 1.** Cytokine concentrations and Seahorse measurements.

**Figure supplement 1.** Expression of virus response genes in β-glucan-trained alveolar macrophages (AMs).

β-glucan and after 7 days AMs were cultured and stimulated ex vivo with LPS (*Figure 4—figure supplement 2A*). Similar to wild-type AMs, the production of pro-inflammatory cytokines CXCL1 and TNF was increased in β-glucan-trained Dectin1-deficient AMs after LPS stimulation (*Figure 4—figure supplement 2B*). This indicates that β-glucan-induced AMs reprogramming is *Clec7a*-independent and that AMs are trained via a different signalling pathway. Since we observed type I IFN gene expression is increased in AMs after β-glucan treatment (*Figure 3E*) and type I IFN was previously reported to be involved in AMs training (*Zahalka et al., 2022*), we next assessed the impact of β-glucan on AMs reprogramming in *Ifnar*$^{-/-}$ mice, which lack type I IFN signalling (*Figure 4—figure supplement 3A*). Similar to Dectin-1-deficient mice, increased production of pro-inflammatory cytokines in response to LPS by β-glucan-treated *Ifnar*$^{-/-}$ AMs was maintained, demonstrating that β-glucan-induced AMs reprogramming is type I IFN independent (*Figure 4—figure supplement 3B*).

Considering recent studies suggesting type II IFN can train AMs following BCG (*Tran et al., 2024*) or adenoviral infection (*Yao et al., 2018*), using *IfngR*$^{-/-}$ mice we next investigated if type II IFN is involved in β-glucan-induced exacerbation of ALI (*Figure 4A*). In contrast to *Dectin1*$^{-/-}$ or *Ifnar*$^{-/-}$ mice, the increased neutrophils recruitment, inflammatory cytokines production, and ALI were abolished in β-glucan-trained *IfngR*$^{-/-}$ mice (*Figure 4B–D*). To investigate the role of type II IFN signalling in β-glucan-induced reprogramming of AMs, we performed ex vivo LPS stimulation on β-glucan-trained *IfngR*$^{-/-}$ AMs (*Figure 4E*). The increase of inflammatory cytokines in response to LPS by β-glucan-trained AMs was abolished (*Figure 4F*), suggesting that training of AM by β-glucan is IFNγ dependent.

Considering the concentration of IFNγ was significantly increased in BAL at day 1 post- β-glucan administration, which was associated with the increased number of neutrophils (*Figure 4G–J*), we next assess the direct role of IFNγ in training of AMs by β-glucan. We selectively depleted IFNγ prior to β-glucan administration in vivo (*Figure 4K*), and found the production of inflammatory cytokines (CXCL1 and TNF) was significantly reduced in β-glucan-trained AMs. Similarly, the depletion of neutrophils prior to β-glucan treatment in mice resulted in significant reduction in inflammatory cytokines in β-glucan-trained AMs (*Figure 4L*). These findings indicate that β-glucan reprograms AMs via IFNγ- and neutrophil-dependent manner.

## Discussion

The evolving field of trained immunity has challenged the boundaries we once drew to discern between the innate and adaptive immune systems. The discovery that the innate immune system can be 'trained' to retain memory-like features has provided novel avenues for prophylactic and therapeutic strategies. The use of an adjuvant like β-glucan has been investigated as an anti-infection agent in acute infections or inflammatory conditions (*Moorlag et al., 2020*; *Zervopoulou et al., 2024*) as well as an anti-cancer treatment (*Vuscan et al., 2024*; *Kalafati et al., 2020*). While we and others have shown that β-glucan can induce central trained immunity by functionally reprogramming the HSC compartment in the bone marrow, the present study highlights an additional facet of β-glucan's immunomodulatory effects on residential immune cells. Here we show that a single intraperitoneal (i.p.) injection of β-glucan can induce trained immunity in the lung AMs and functionally reprograms AMs at both the transcriptional and metabolic levels. These findings underscore the occurrence of tissue-specific immune training following systemic treatment, demonstrating processes of β-glucan-mediated peripheral trained immunity occurring in parallel with central immunity. We subsequently demonstrate that β-glucan-trained AMs can be detrimental after LPS or poly(I:C) administration causing severe ALI. Thus, this study is shedding light on the deleterious effects of trained immunity and its impact on immunopathology, which is incompletely understood.

There are several key features in trained immunity, including shifts in epigenetic and transcriptomic profiles after initial stimulation that returns to steady states. However, this epigenetic and metabolic remodelling result in long-term memory with heightened responses to diverse secondary stimulation (*Divangahi et al., 2021*). The nature of each stimulus appears to be critical in determining the

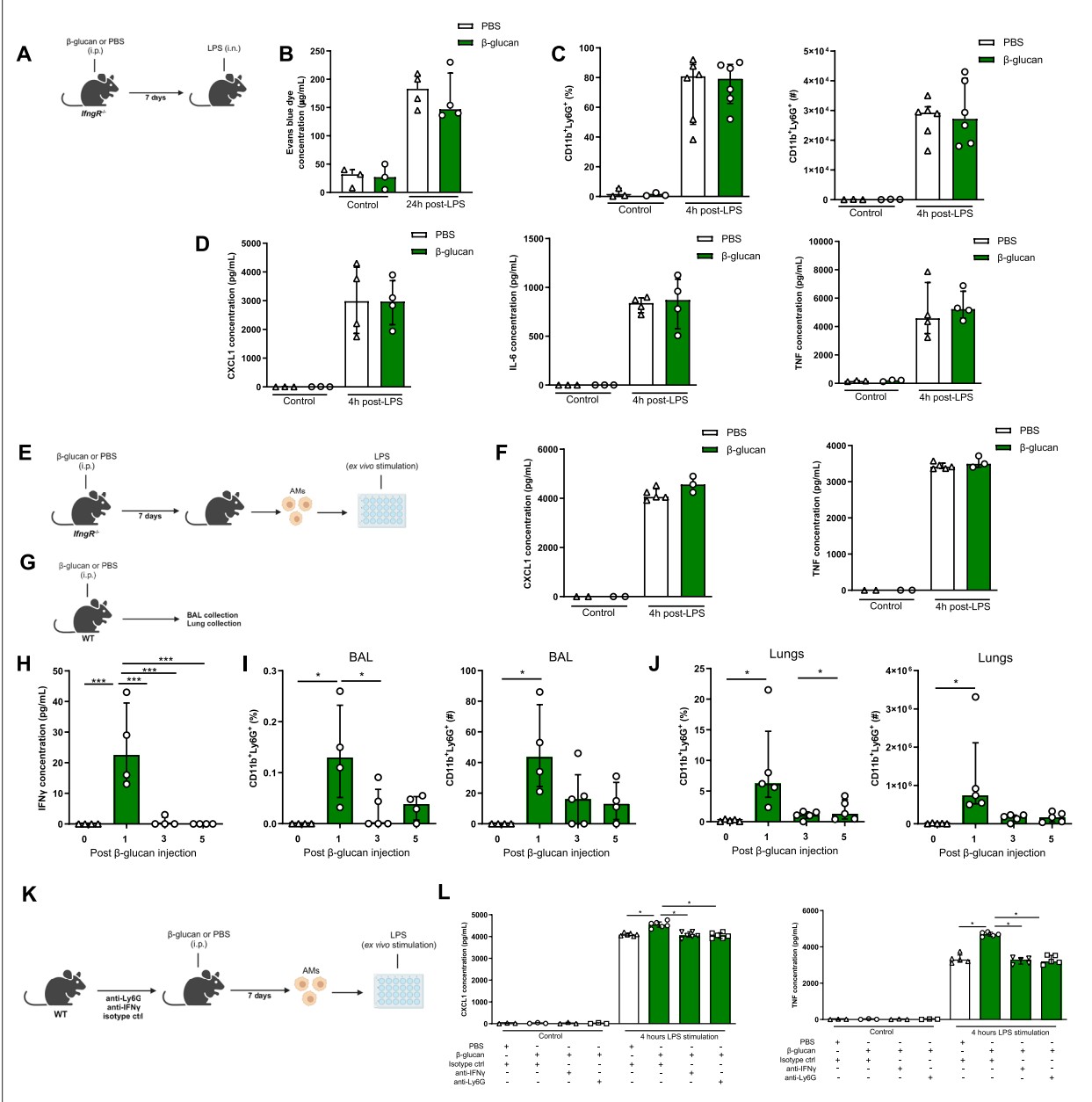

**Figure 4.** IFNγ and neutrophils are required in β-glucan-mediated alveolar macrophage (AM) reprogramming. (**A**) Schematic of the β-glucan-induced training and lipopolysaccharide (LPS)-induced acute lung injury (ALI) model. Experiments were performed in sex- and age-matched 10- to 12-week-old control (i.p. PBS, white bars) and trained (i.p. β-glucan, green bars) *IfngR⁻/⁻* mice. (**B**) Alveolar capillary membrane permeability assessed by lung Evans blue dye concentration. (**C**) Quantification of bronchoalveolar lavage (BAL) neutrophils proportion (left) and absolute count (right) (gated on single live cells, CD45.2⁺, CD11c⁻, Siglec-F⁻, CD11b⁺, Ly6G⁺). (**D**) BAL chemokine C-X-C motif ligand 1 (CXCL1) concentration (left) and pro-inflammatory cytokines (IL-6: interleukin-6 (middle) and TNF: tumour necrosis factor (right)) concentrations. (**E**) Schematic of control (i.p. PBS, white bars) or β-glucan-trained (i.p. β-glucan, green bars) AMs collected from adult *IfngR⁻/⁻* mice ex vivo stimulation with LPS. (**F**) Chemokine C-X-C motif ligand 1 (CXCL1) and tumour necrosis factor (TNF) concentrations after ex vivo LPS stimulation. (**G**) Schematic of the analysis of the effect of i.p. β-glucan injection on interferon-γ (IFNγ) production and neutrophils expansion before (white bars) and days 1, 3, 5, and 7 post-injection (green bars). (**H**) BAL IFNγ concentrations. (**I**) Quantification of BAL neutrophils proportion (left) and absolute count (right) (gated on single live cells, CD45.2⁺, CD11c⁻, Siglec-F⁻, CD11b⁺, Ly6G⁺). (**J**) Quantification of lung neutrophils proportion (left) and absolute count (right). (**K**) Schematic of control (i.p. PBS, white bars) or β-glucan-trained (i.p. β-glucan, green bars) AMs ex vivo stimulation with 50 ng/ml LPS. AMs were collected from control (i.p. injection of isotypes), neutrophils depleted (i.p. injection of anti-Ly6G antibodies), or IFNγ antibody-depleted (i.p. injection of anti-IFNγ antibodies) adult WT mice. (**L**) CXCL1 and TNF concentrations after ex vivo LPS stimulation. Data were analysed using one-way ANOVA followed by Dunn's multiple comparisons test. *p < 0.05, ***p < 0.001. Schematics created using BioRender.com.

*Figure 4 continued on next page*

*Figure 4 continued*

The online version of this article includes the following source data and figure supplement(s) for figure 4:

**Source data 1.** Individual measurements, cytokine concentrations, cell frequencies and cell numbers.

**Figure supplement 1.** β-Glucan-aggravated acute lung injury (ALI) is independent of Dectin-1.

**Figure supplement 1—source data 1.** Cytokine concentrations, cell frequencies and cell numbers.

**Figure supplement 2.** β-Glucan-mediated alveolar macrophage (AM) reprogramming is independent of Dectin-1.

**Figure supplement 2—source data 1.** Cytokine concentrations.

**Figure supplement 3.** β-Glucan-mediated alveolar macrophage (AM) reprogramming is independent of type I interferon signalling.

**Figure supplement 3—source data 1.** Cytokine concentrations.

---

immunological outcome. While we show here that β-glucan aggravates LPS-induced immunopathology, repeated challenges of LPS has also long been associated with a dampening of immune responses, termed tolerance (*Freudenberg and Galanos, 1988*; *Mason et al., 1997*). However, ambient exposure of LPS has also been linked with training leading to enhanced bacterial clearance (*Zahalka et al., 2022*). Evidently, the impacts of trained immunity are intricate and context dependent. We postulate two signals are required for the induction of trained immunity: signal 1, which is essential for reprogramming innate immune cells centrally (HSCs) or peripherally (e.g., AMs), and signal 2, an environmental cue critical for activating their functional capacity. In this study, we provide evidence supporting this conceptual framework.

β-Glucan-trained AMs were able to generate a robust response both in vitro and in vivo to bacterial (LPS) or viral (poly(I:C)) ligands. The increased response in β-glucan-trained AM was maintained up to 28 days, which is an indication of long-term reprogramming of AMs. The reprogramming of trained AMs was supported by significant alterations in both the transcriptomic and metabolic states. Using adoptive transfer experiments, we then showed that the functional reprogramming of β-glucan-trained AMs persisted even in the absence of the initial systemic β-glucan administration. This finding highlights the persistence of the intrinsic training program in AMs.

AMs are originated from yolk sac and fetal liver monocytes, which seeds the lung as soon as we take our first breath of air (*Guilliams et al., 2013*). Importantly, AMs are the first immune cell to response to any particles or pathogens that reach the lower airways of the lungs, thus alterations in their functional capacity can significantly impact subsequent immune responses. For instance, given AMs are in direct contact with surfactant (produced by alveolar type 2 epithelial cells) and invading pathogens, a defect in GM-CSF or TGFβ signalling leads to accumulation of surfactant (alveolar proteinosis) and increased susceptibility to pulmonary infections (*Baker et al., 2010*; *Huffman et al., 1996*). AMs are able to sense and integrate multiple environmental signals – such as pH (*Huffman et al., 1996*), temperature (*Kashio et al., 2012*; *Link et al., 2010*), osmolarity (*Fang et al., 2009*), metabolites including fatty acids (*Haldar et al., 2020*; *Okabe and Medzhitov, 2014*), extracellular membrane components, and danger signals (*Sun et al., 2013*) – to maintain tissue homeostasis (*Okabe and Medzhitov, 2016*). However, the mechanisms underlying their ability to adapt to environmental stimuli while maintaining lung tissue homeostasis without impairing gas exchange is largely unknown (*Lazarov et al., 2023*). AMs exhibit a remarkable plasticity, demonstrating a spectrum of functional polarization that ranges from regulatory to pro- and anti-inflammatory states. For instance, AM anti-inflammatory polarization in a LPS model of sepsis was TNF dependent, as AMs exposed to TNF exhibited diminished phagocytosis, superoxide anion ($O_2^-$) and CXCL1 production, with reduced neutrophils recruitment (*Mason et al., 1997*). Consequently, these mice had a reduction in lung clearance of *P. aeruginosa* infection. Influenza virus was able to induce similar anti-inflammatory function in AMs with decreased CXCL1 production and neutrophils recruitment via type I IFN pathway (*Shirey et al., 2019*), leading to an increased susceptibility to super bacterial infection. On the other hand, AMs appear to be more resistant to polarization towards a pro-inflammatory state (*Tomlinson et al., 2012*). A recent study demonstrated that exposure of lungs to ambient amount of LPS trains AMs in type I IFN dependent, but type II IFN and T cell-independent manner (*Zahalka et al., 2022*). However, in our model system, systemic administration of β-glucan trains AMs in a type II IFN dependent, but type I IFN-independent manner. Similarly, in live infection models, we and others have identified IFNγ signalling a key player in AM training after BCG vaccination (*Tran et al., 2024*), influenza infection (*Wang et al., 2023*), pneumococcal infection (*Arafa*

*et al., 2022*), and intranasal infection with an adenoviral vector (*Zhou and Liao, 2021*). Although we have not identified which cells produce IFNγ in the β-glucan model, it has been demonstrated by our group and others that following BCG vaccination, CD4+ T cells are the major source of IFNγ (*Tran et al., 2024*), whereas after pulmonary adenovirus infection, CD8+ T cells predominantly produce IFNγ (*Yao et al., 2018*). Influenza infection has also been described to induce IFNγ-dependent AM training with NK cells being the major source (*Wang et al., 2023*). Interestingly, we found that neutrophils were also required for β-glucan-mediated AMs training. We have recently shown that β-glucan can reprogram HSCs via Dectin 1 and type I IFN signalling to promote granulopoiesis and the generation of trained neutrophils promoting disease tolerance to influenza infection (*Khan et al., 2025*). Although we have shown that the recruitment of these trained neutrophils and type II IFN signalling were required for training AMs, the cellular and molecular mechanisms involved in this dialogue is unknown and requires further investigation. Interestingly, we have recently demonstrated that, in addition to GM-CSF and TGFβ signalling, neutrophils are critical for AMs self-renewal and maintenance during early lung development via the production of 12-hydroxyeicosatetraenoic acid (12-HETE) (*Pernet et al., 2023*). The absence of 12-HETE leads to a significant reduction in the number AMs in adult lungs, enhanced senescence, and consequently increased susceptibility to IAV or SARS-CoV-2 infection. Thus, there might be a constant bidirectional dialogue between neutrophils and AMs, with neutrophils providing cues from internal organelles to AMs, and AMs offering signals from the external environment to neutrophils, which then return to their graveyard in the BM.

Although we have not directly tested the contribution of circulating monocytes in the initial reprogramming of AMs via β-glucan, the persistence of trained immunity by adoptively transferred AMs into *Csf2rb*⁻/⁻ mice suggests that the maintenance of the trained AM state was independent of bone marrow-derived monocytes. Additionally, the findings from Theobald et al. indicate that Dectin-1 is the receptor responsible for recognizing β-glucan which has been shown to activate macrophages and induce trained immunity in several models (*Khan et al., 2025*; *Cheng et al., 2014*; *Arts et al., 2016*; *Quintin et al., 2012*). Here we found that β-glucan-mediated AM training, when administered systemically, was independent of Dectin-1. There are two potential explanations for this observation. First, it has been shown that β-glucan can also activate signalling via other TLRs, particularly TLR2 or complement receptor 3 (CR3) (*Yadav and Schorey, 2006*; *Thornton et al., 1996*; *Xia et al., 1999*), and second as the β-glucan is a particulate, its internalization by phagocytes can also initiate signalling (*McLeish et al., 1998*; *Yang et al., 2011*; *Qi et al., 2011*). Additionally, our understanding of how administration of β-glucan in peritoneal cavity leads to HSCs training in the BM and AMs training in thoracic cavity is extremely limited. Thus, addressing the molecular mechanisms of β-glucan signalling pathways (e.g., Dectin1 dependent and independent) in both immune and non-immune cells, as well as its mode of action (e.g., direct access to an organ versus indirect effects via systemic release of cytokines), is necessary to delineate the deleterious versus protective effects of β-glucan-mediated trained immunity.

β-Glucan is present in the cell wall of all fungi but will vary between different species and strains. In fact, the differential 1,3-1,6 glycosidic branching and molecular weight significantly impacts the response to the compound with a large variability of scientific findings contingent on the type of β-glucan used in a study (*Luo et al., 2019*; *Moerings et al., 2021*; *Akramiene et al., 2007*). Fungi make up a portion of the human microbiome, termed the mycobiome. Several studies have described a high mycotic diversity between different populations, and even significant variability within an individual overtime (*Sun et al., 2021*; *Szóstak et al., 2023*; *Nash et al., 2017*). The variance in gut colonization by fungal species can moreover cause gut dysbiosis which has been associated with poorer outcomes during SARS-CoV-2 infection, sepsis and cancer immunotherapy (*Reinold et al., 2022*; *Chen et al., 2022*; *Huang et al., 2023*; *Orieux et al., 2023*; *Prevel et al., 2022*). It is tempting to postulate that the levels of the mycobiome as well as its composition can influence subsequent immune reactions partly due to a distinct β-glucan makeup. In fact, antibodies to various types of β-glucan was detected in adult sera with different levels correlating with a person's occupation (*Noss et al., 2012*). Thus, the quantity and quality of circulating β-glucan in an individual at steady state can remarkably affect the subsequent immune responses to sterile or microbial inflammation. The concept of detrimental training by endogenous agents has also been described with heme, which similarly aggravated LPS-induced inflammation (*Jentho et al., 2021*.) Basal levels of such immunomodulatory molecules provide a basis for host response heterogeneity and corresponding excessive inflammation

(*Grasselli et al., 2020*). Understanding these underlying processes may provide important insights for developing novel therapeutic approaches against immune-mediated pulmonary disease.

# Materials and methods

## Key resources table

| Reagent type (species) or resource | Designation | Source or reference | Identifiers | Additional information |
|---|---|---|---|---|
| Strain, strain background (*Mus musculus*) | C57BL/6 | Jackson Laboratories | Strain #: 000664; RRID:IMSR_JAX:000664 | |
| Strain, strain background (*M. musculus*) | Csf2rb⁻/⁻ | Jackson Laboratories | Strain #: 005940; RRID:IMSR_JAX:005940 | |
| Strain, strain background (*M. musculus*) | Clec7a (Dectin1)⁻/⁻ | Jackson Laboratories | Strain #: 012337; RRID:IMSR_JAX:012337 | |
| Strain, strain background (*M. musculus*) | Ifnar⁻/⁻ | Jackson Laboratories | Strain #: 028288; RRID:IMSR_JAX:028288 | |
| Strain, strain background (*Mus musculus*) | IfngR⁻/⁻ | Jackson Laboratories | Strain #: 003288; RRID:IMSR_JAX:003288 | |
| Commercial assay or kit | Mouse CXCL1 DuoSet ELISA | R&D Systems | Cat #: DY453 | |
| Commercial assay or kit | Mouse TNF-alpha DuoSet ELISA | R&D Systems | Cat #: DY410 | |
| Commercial assay or kit | Mouse IL-6 DuoSet ELISA | R&D Systems | Cat #: DY406 | |
| Commercial assay or kit | Mouse IFNγ DuoSet ELISA | R&D Systems | Cat #: DY485 | |
| Commercial assay or kit | Pierce BCA assay | Thermo Fisher | Cat #: 23225 | |
| Commercial assay or kit | Seahorse XF Cell Mito Stress Test Kit | Agilent | Cat #: 103015-100 | |
| Antibody | Fixable Viability Dye eFluor 506 | Invitrogen | Cat #: 65-0866-14 | FACS (1:1000) |
| Antibody | Purified Rat Anti-Mouse CD16/CD32 (Mouse BD Fc Block) | BD Bioscience | Cat #: 553141; RRID:AB_394656 | FACS (1:200) |
| Antibody | PE-Cy7-conjugated anti-CD11c (Mouse monoclonal) | BD Bioscience | Cat #: 561022; RRID:AB_647251 | FACS (1:200) |
| Antibody | BV786-conjugated anti-Siglec-F (Mouse monoclonal) | BD Bioscience | Cat #: 740956; RRID:AB_2740581 | FACS (1:200) |
| Antibody | BUV395-conjugated anti-CD11b (Mouse monoclonal) | BD Bioscience | Cat #: 565976; RRID:AB_2738276 | FACS (1:200) |
| Antibody | APC-Cy7-conjugated anti-Ly6G (Mouse monoclonal) | BD Bioscience | Cat #: 560600; RRID:AB_1727561 | FACS (1:200) |
| Antibody | BUV737-conjugated anti-CD45.2 (Mouse monoclonal) | BD Bioscience | Cat #: 612778; RRID:AB_2870107 | FACS (1:200) |
| Antibody | PE-conjugated anti-IFNγ (Mouse monoclonal) | BD Bioscience | Cat #: 554412; RRID:AB_395376 | FACS (1:200) |
| Antibody | FITC-conjugated anti-CD45.2 (Mouse monoclonal) | BD Bioscience | Cat #: 553772; RRID:AB_395041 | In vivo (2 µg per mouse) |
| Antibody | BUV395-conjugated anti-CD45.2 (Mouse monoclonal) | BD Bioscience | Cat #: 564616; RRID:AB_2738867 | FACS (1:200) |
| Antibody | Depleting anti-IFNγ (rat IgG1k) (Mouse monoclonal) | Biolegend | Cat #: 505801; RRID:AB_315395 | In vivo (200 µg per mouse) |
| Antibody | Depleting anti-Ly6G (rat IgG2a,k) (Mouse monoclonal) | Biolegend | Cat #: 127601; RRID:AB_1089179 | In vivo (70 µl per mouse) |

*Continued on next page*

*Continued*

| Reagent type (species) or resource | Designation | Source or reference | Identifiers | Additional information |
|---|---|---|---|---|
| Chemical compound | Clodronate liposomes | Liposoma BV | Cat #: C-005 | In vivo (200 µg per mouse) |
| Other | *Escherichia coli* O55:B55 LPS | Sigma-Aldrich | Cat #: L2637 | In vivo (50 µg per mouse) |
| Other | β-1,3-Glucan purified from *Saccharomyces cerevisiae* | Sigma-Aldrich | Cat #: G5011 | In vivo (1 mg per mouse) |
| Other | Poly(I:C) HMW | Invivogen | Cat #: tlrl-pic | In vivo (50 µg per mouse) |

## Mice

C57BL/6, *Csf2rb*$^{-/-}$, *Clec7a*$^{-/-}$, *Ifnar*$^{-/-}$, and *IfngR*$^{-/-}$ mice were purchased from Jackson Laboratories. All animals were housed and inbred at the animal facility of the Research Institute of McGill University under specific pathogen-free conditions with ad libitum access to food and water, a temperature of 21°C (±1°C), relative humidity of 40–60% (±5%), and light cycle of 12 hr on, 12 hoff (daily cycle). *Sex as a biological variable*: Mice were randomly allocated to experimental groups, and experiments were performed using both female and male age- and sex-matched mice. Similar findings were reported for both sexes.

## β-Glucan training

Mice were administered intraperitoneally with 1 mg of β-1,3-glucan purified from *Saccharomyces cerevisiae* (Sigma) diluted in 100 µl of PBS 7 or 28 days before lung injury or AMs collection for ex vivo stimulation.

## ALI models

Mice were administered with 50 µg of *Escherichia coli* O55:B55 LPS (Sigma) or poly(I:C) (Invivogen) in PBS (25 µl per mouse, intranasally) to induce TLR-4- or 3-mediated ALI, respectively.

## Lung microCT scan

The trachea was cannulated with a 22-gauge cannula and an intra-thoracic pressure of 25 mmH$_2$O was generated using a manometer. Images were acquired right after lung inflation using the nanoScan SPECT + CT (Mediso) allowing a resolution of 20 µm. DICOM software was used to analyse the microCT scans determining the average lung Hounsfield unit (HU) and the proportion of non- or poorly aerated lung (HU –500; +100).

## BAL and lung collection

Broncho-alveolar lavage (BAL) samples collected by cannulating the trachea with a 22-gauge cannula, then washing the lungs with 3× 1 ml of cold, sterile PBS. The total volume of the recovered fluid after lavage was around 0.7 ml. Samples were centrifuged (1500 rpm, 10 min). Lung tissues were perfused with 10 ml of PBS, collected and minced before collagenase type I (3 mg, Worthington CLS-1), elastase (3 mg, Worthington ESL), and DNase I (0.4 mg, Worthington D) digestion for 30 min at 37°C. Lungs were filtered through a 70-µm nylon mesh, and red blood cells were lysed.

## Endothelial permeability

LPS or poly(I:C)-challenged mice were intravenously injected with 400 µl of Evan's blue dye (2% in PBS) into the mice. After 1 hr, mice were euthanized, and lungs were perfused with 10 ml of PBS. Evan's blue then extracted by overnight incubation in formamide at 56°C (lungs) and quantified by spectrophotometry analysis using a standard curve of Evan's blue in formamide.

## Histopathological analysis

Histopathological analysis was performed as previously described (*Pernet et al., 2023*). Lungs were inflated and fixed for 48 hr with 10% formalin, and then embedded in paraffin. Sections (5 µm) were cut and stained with haematoxylin and eosin. Slides were scanned at a resolution of ×40 magnification,

and pictures were taken using a Leica Aperio slide scanner (Leica). Histology samples were evaluated according to ATS 2011 guidelines regarding 'Features and measurements of experimental ALI in animals' by a blinded observer.

## ELISA

CXCL1, TNF-α, IL-6, and IFNγ levels in BAL were assessed by ELISA (R&D Systems).

## Protein in BAL

Samples were centrifuged (1500 rpm, 10 min), and total protein content was assessed using a Pierce BCA Protein assay (Thermo Fisher).

## Flow cytometry

BAL and total lung cell counts were determined with a haemocytometer, and 1–2 illion cells were used for staining. Cells were initially stained with viability dye e506 (Invitrogen, 20 min, 4°C) and surface stained with anti-CD16/32 (BD Bioscience) in 0.5% BSA/PBS solution to block nonspecific AB interaction with Fc receptors (10 min, 4°C). Cells were then surface-stained with different combinations of PE-Cy7-conjugated anti-CD11c, BUV786-conjugated anti-Siglec-F, BUV395-conjugated anti-CD11b, APC-Cy7-conjugated anti-Ly6G, and BUV737-conjugated anti-CD45.2 antibodies (all from BD Biosciences). For IFNγ intra-cellular staining, cells were fixed and permeabilized using BD CytoFix/CytoPerm (BD Bioscience) before intracellular staining with PE-conjugated anti-IFNγ antibodies (BD Biosciences). Flow cytometry was performed using a BD LSR Fortessa X-20 instrument (BD Biosciences) with FACSDiva software v.8.0.1 (BD Biosciences). Analysis was performed using FlowJo software v.10.7.1 (Tree Star).

## Intravascular staining

In vivo discrimination between pulmonary vasculature and parenchyma was performed as previously described (*Pernet et al., 2023*). Adult WT mice were given 2 µg of FITC-conjugated anti-CD45.2 intravenously to label all circulating cells. Three minutes later, mice were euthanized and lungs collected, stained ex vivo with BUV395-conjugated anti-CD45.2 antibody to determine the parenchymal (cells only labelled with the ex vivo antibody) or vascular localization of the cells (cells labelled with both antibodies).

## AM depletion

WT mice were treated with control or clodronate liposomes (70 µl, intranasally; Liposoma BV). The LPS-induced ALI was then performed at day 2 after clodronate instillation.

## Adoptive transfer models

AMs from WT mice which received i.p. PBS or β-glucan were collected as described above and resuspended at a density of $5 \times 10^4$ cells per 5–7 µl of RPMI1640 medium supplemented with 10% (vol/vol) FBS, 2 mM L-glutamine, 10 mM HEPES, and 100 U ml$^{-1}$ penicillin–streptomycin. AMs were then transferred by the intranasal route into day 2 *Csf2rb*$^{-/-}$ pups. LPS-induced ALI was performed 6 weeks after AM adoptive transfer. BAL and lung tissue were collected and processed as described above for endothelial permeability, flow cytometry, total BAL cytokine, and protein content experiments.

## Isolation and culture of AMs

AMs were collected by BAL of naive mice using cold, sterile PBS ($5 \times 1$ ml for adult mice). AMs were cultured in the specific media described above. After 1 hr of adhesion, AMs were washed with PBS and placed in fresh medium.

## Ex vivo stimulation

AMs from WT, *Dectin1*$^{-/-}$, *Ifnar*$^{-/-}$, and *IfngR*$^{-/-}$ mice which received i.p. PBS or β-glucan were collected as described above and $5 \times 10^4$ cells in specific media were distributed per well. AMs were stimulated with 50 ng/ml of LPS (Sigma) for 4 hr at 37°C.

## Library preparation and RNA-seq

Total RNA was collected from BAL AMs from four WT mice per conditions (i.p PBS, no ex vivo stimulation/i.p. β-glucan, no ex vivo stimulation/i.p PBS, ex vivo LPS stimulation/i.p. β-glucan, ex vivo LPS stimulation). After RNA quality controls, sequencing libraries were constructed using the Illumina TruSeq protocol. Libraries were sequenced on an Illumina NovaSeq 6000 (paired-end 100 base pair) to an average depth of 51,189,336 reads per sample.

## RNA-seq data analysis

RNA-seq reads were aligned to the *Mus musculus* genome from Ensembl version 99 using STAR (version 2.7.3a) was used (*Dobin et al., 2013*). All regions overlapped between referenced exons and alignments were counted using featureCounts (subread-1.6.4) (*Liao et al., 2014*). Low abundance genes were filtered out leaving 12,894 genes for subsequent analysis.

Differential expression analyses were performed using the DESeq2 package (DESeq2 1.40.2) (*Love et al., 2014*). Gene ontology analysis was realized with the R package TopGO. For GSEA a ranked list of the differentially expressed genes was used with clusterProfiler v4.8.2 (*Wu et al., 2021*), and the Molecular Signatures Database MSigDB v7.5.1.

## Extracellular flux analysis

Seahorse assay of isolated cells was performed as previously described (*Pernet et al., 2023*). Real-time OCRs of AMs were measured in XF medium (non-buffered RPMI containing 2 mM L-glutamine, 25 mM glucose, and 1 mM sodium pyruvate) using a Seahorse Xfe 96 Analyzer (Agilent Technologies). For the mitochondrial stress test, mitochondrial inhibitors oligomycin (1.5 μM), fluorocarbonyl cyanide phenylhydrazone (FCCP) (1 μM), antimycin A, and rotenone (0.5 μM) were used as per the manufacturer's recommendations. In brief, cells were seeded at a density of 100,000 cells per well and 3 basal measurements were taken. Following this, two consecutive measurements were taken following each injection of oligomycin, FCCP, and antimycin A with rotenone. All measurements were normalized to cell number using crystal violet dye extraction assay. Oxygen consumption curves, OCRs and ECARs were generated using Wave Desktop 2.3 (Agilent Technologies).

## IFNγ and neutrophils depletion experiments

Sex- and age-matched adult WT mice received intra-peritoneal 200 μg of anti-IFNγ (rat IgG1k, Biolegend), anti-Ly6G (rat IgG2a,k, Biolegend) or appropriated control isotypes injection at days 1, 0, 2, 4, and 6 per intraperitoneal injection of β-glucan. AMs collection for ex vivo stimulation was performed at day 7 after intraperitoneal β-glucan injection.

## Statistical analysis

No statistical methods were used to pre-determine samples sizes. Sample sizes were empirically determined to optimize numbers based on previous experience with equivalent experiments reported in previous publications (*Tran et al., 2024*; *Pernet et al., 2023*). A minimum sample size of 3 was always included for statistical analysis to be valid. Data distribution was assumed to be normal, but this was not formally tested. No data points were excluded from analysis. Key experiments were reproduced independently two times for reproducibility of findings except RNA-seq as we considered the high output of data to be sufficient to corroborate our other reproducible findings. Experimentally intricate experiments such as Ly6G/IFNγ depletion and neonatal adoptive transfer was performed once. All experiments involve mice or mouse-derived samples. Mice of the same sex were age-matched and randomly assigned prior to the initiation of experiments. Data collection and analysis were not performed blinded to the experimental conditions as experiments were all done by one researcher at a time. Statistical analyses were performed using GraphPad Prism v.9.1.2 software (GraphPad). Statistical differences were determined using one-way ANOVA followed by Dunn's multiple comparisons test, paired or unpaired two-tailed $t$-test or two-tailed Mann–Whitney test. Differential gene expression analysis was carried out using DEseq2 package (*Love et al., 2014*).

## Acknowledgements

The authors acknowledge technical help from staff at the RI-MUHC histopathology platform and RI-MUHC Small Animal Imaging Laboratory. M.D. is funded by Canadian Institute of Health Research (CIHR) Project Grant-168885 and MM1174910, a Fonds de recherche du Québec Santé (FRQS) Award, holds the Strauss Chair in Respiratory Diseases and is a fellow member of the Royal Society of Canada. E.P. and R.P. are fellows supported by a Postdoctoral Fellowship from the Fonds de Recherche du Québec Santé.

## Additional information

### Funding

| Funder | Grant reference number | Author |
| --- | --- | --- |
| Canadian Institutes of Health Research | 168885 | Maziar Divangahi |
| Canadian Institutes of Health Research | MM1174910 | Maziar Divangahi |
| Fonds de Recherche du Québec - Santé | Award | Maziar Divangahi |
| Fonds de Recherche du Québec - Santé | Postdoctoral Fellowship | Erwan Pernet Renaud Prevel |
| Royal Society of Canada | | Maziar Divangahi |

The funders had no role in study design, data collection, and interpretation, or the decision to submit the work for publication.

### Author contributions

Renaud Prevel, Conceptualization, Data curation, Formal analysis, Investigation, Writing – original draft; Erwan Pernet, Conceptualization, Investigation, Methodology; Kim A Tran, Data curation, Formal analysis, Investigation, Writing – original draft, Writing – review and editing; Abderrahmane Sadek, Data curation, Formal analysis; Mina Sadeghi, Elizabeth Lapshina, Leonardo F Jurado, Data curation; Arnold S Kristof, Investigation; Mohieddine Moumni, Data curation, Investigation; Jeremie Poschmann, Data curation, Formal analysis, Investigation; Maziar Divangahi, Conceptualization, Supervision, Funding acquisition, Investigation, Writing – review and editing

### Author ORCIDs

Kim A Tran https://orcid.org/0000-0003-0299-7417
Jeremie Poschmann https://orcid.org/0000-0002-9613-5297
Maziar Divangahi https://orcid.org/0000-0002-1254-6110

### Ethics

All experiments involving animals were approved by the McGill University Animal Care Committee (permit number 2010-5860) in accordance with the guidelines set out by the Canadian Council on Animal Care.

Reviewer #1 (Public review): https://doi.org/10.7554/eLife.102068.3.sa1
Reviewer #2 (Public review): https://doi.org/10.7554/eLife.102068.3.sa2
Author response https://doi.org/10.7554/eLife.102068.3.sa3

## Additional files

### Supplementary files

MDAR checklist

## Data availability

All data supporting the findings of this study are included in the published article and supplementary materials. Bulk RNA-seq data have been deposited to the European Nucleotide Archive and can accessed on https://www.ebi.ac.uk/ena/browser/home with accession code PRJEB75517 and secondary accession code ERP160094. Source data are provided with this paper.

The following dataset was generated:

| Author(s) | Year | Dataset title | Dataset URL | Database and Identifier |
|---|---|---|---|---|
| Prével R, Pernet E, Tran KA, Sadek A, Sadeghi M, Lapshina E, Jurado L, Kristof AS, Moumni M, Poschmann J, Divangahi M | 2025 | β-glucan enhances LPS-induced acute lung injury via interferon γ-mediated alveolar macrophages reprogramming | https://www.ebi.ac.uk/ena/browser/view/PRJEB75517 | EBI European Nucleotide Archive, PRJEB75517 |

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
