## [Editor Report · eLife Assessment]

This **important** study advances our understanding of maladaptive innate immune training. The experimental evidence supporting the conclusions is **convincing** and the expert reviewers strongly endorse the manuscript. The work will be of high interest to both researchers in the trained immunity field and clinician scientists.

---

## [Referee Report · Reviewer #1 (Public review)]

Summary:

The concept that trained immunity, as defined, can be beneficial to subsequent immune challenges is important in the broad context of health and disease. The significance of this manuscript is the finding that trained immunity is actually a two-edged sword, herein, detrimental in the context of LPS-induced Acute Lung Injury that is mediated by AMs.

Strengths:

Several lines of evidence in different mouse models support this conclusion. The postulation that differences in immune responses in individuals is linked to differences in the mycobiome and consequent B-glucan makeup is provocative.

Weaknesses:

However, the findings that the authors state are relevant to sepsis are actually confined to a specific lung injury model and not classically-defined sepsis, the ontogeny of the reprogrammed AMs is uncertain, and links in the proposed signaling pathways need to be strengthened.

Comments on the latest version:

The manuscript is improved with further clarifications and additional experimentation. My prior concerns are addressed.

---

## [Referee Report · Reviewer #2 (Public review)]

Summary:

Prével et al. present an in vivo study in which they reveal an interesting aspect of β-glucan, a known inducer of enhanced immune responses termed trained immunity in sterile inflammation. The authors can show that β-glucan's can reprogram alveolar macrophages (AMs) in the lungs through neutrophils and IFNγ signaling and independent of Dectin1. This reprogramming occurs at both transcriptional and metabolic levels. After β-glucan training, LPS induced sterile inflammation exacerbated acute lung injury via enhanced immunopathology. These findings highlight a new aspect of β-glucan's role in trained immunity and its potential detrimental effects when enhanced pathogen clearance is not required.

Strengths:

- This manuscript is well-written and effectively conveys its message.

- The authors provide important evidence that β-glucan training is not solely beneficial but depending on the context can also enhance immunopathology. This will be important to the field for two reasons. It shows again that trained immunity can also be harmful. Jentho et al. 2021 had already provided further evidence for this aspect. And it highlights anew that LPS application is an insufficient infection model.

Original weaknesses noted:

- Only a little physiological data from the in vivo models is provided.

- Effects in histology appear to be rather weak.

Comments on latest version:

The authors have revised the new version according to my suggestions or responded in a sufficient manner to my requests, with one exception. I recommend to rename TNF as explained by Grimstad in JAMA Dermatol. 2016;152(5):557.

---

## [Author Response]

The following is the authors’ response to the original reviews

**Reviewer #1 (Public review):**
Summary:The concept that trained immunity, as defined, can be beneficial to subsequent immune challenges is important in the broad context of health and disease. The significance of this manuscript is the finding that trained immunity is actually a two-edged sword, herein, detrimental in the context of LPS-induced Acute Lung Injury that is mediated by AMs.Strengths:Several lines of evidence in different mouse models support this conclusion. The postulation that differences in immune responses in individuals are linked to differences in the mycobiome and consequent B-glucan makeup is provocative.Weaknesses:The findings that the authors state are relevant to sepsis, are actually confined to a specific lung injury model and not classically-defined sepsis. In addition, the ontogeny of the reprogrammed AMs is uncertain. Links in the proposed signaling pathways need to be strengthened.
**Reviewer #2 (Public review):**
Summary:Prével et al. present an in vivo study in which they reveal an interesting aspect of β-glucan, a known inducer of enhanced immune responses termed trained immunity in sterile inflammation. The authors can show, that β-glucan's can reprogram alveolar macrophages (AMs) in the lungs through neutrophils and IFNγ signaling and independent of Dectin1. This reprogramming occurs at both transcriptional and metabolic levels. After β-glucan training, LPS-induced sterile inflammation exacerbated acute lung injury via enhanced immunopathology. These findings highlight a new aspect of β-glucan's role in trained immunity and its potential detrimental effects when enhanced pathogen clearance is not required.Strengths:(1) This manuscript is well-written and effectively conveys its message.(2) The authors provide important evidence that β-glucan training is not solely beneficial, but depending on the context can also enhance immunopathology. This will be important to the field for two reasons. It shows again, that trained immunity can also be harmful. Jentho et al. 2021 have already provided further evidence for this aspect. And it highlights anew that LPS application is an insufficient infection model.Weaknesses:(1) Only a little physiological data is provided by the in vivo models.(2) The effects in histology appear to be rather weak.
**Reviewer #1 (Recommendations for the authors):**
The opening paragraph in the introduction focuses on sepsis. This is misleading since this manuscript does not address sepsis but rather intranasal-administered LPS-induced acute lung injury.

We are in total agreement with the reviewer and have modified the introduction to focus on acute lung injury with clinical relevance more associated to TLR4-mediated acute lung injury and lung inflammation.

The authors make definitive statements that AMs originate from fetal liver monocytes. However, it is well known that the ontogeny of AMs is complex and AMs can be populated, in part, from peripheral monocytes. The ontogeny of reprogrammed AMs was not addressed in this study but they may come from monocyte-derived AMs following B-glucan training (transfer of AMs into Csf2rb KO mice does not prove the contrary). In this regard, do, for example, the percentages of CD11b+ AMs change? More phenotyping of the control and reprogrammed AMs would enhance the interpretation of the findings.

The reviewer is correct that the ontogeny of AMs can be heterogenous, especially following a pulmonary challenge. In β-glucan-treated mice, Figure 1I shows no changes in frequency or number of AMs in the BAL. As the reviewer suggested, we repeated this experiment and incorporate more markers for AMs. New Supplementary Figure 1C shows the expression of CD11b on AMs (CD11c^+^SiglecF^+^) from control and β-glucan-treated mice. While the frequency increases with LPS administration, we show no difference between control and β-glucan groups suggesting β-glucan does not induce the expansion of monocyte-derived AMs. Additionally, in New Supplementary Figure 1D, we show the expression of AM-associated markers in order to better delineate their phenotype. We observed no differences in MHCII, CD169, CD64 and F4/80 in β-glucan-treated mice, but an increase in CD80^+^ AMs following βglucan suggesting enhanced activation corroborating their proinflammatory phenotype. Collectively, these data indicate that while the frequency and number of either yolk-sac or BMderived AMs are unchanged in the β-glucan treated mice, the activation of AMs is enhanced after the systemic treatment with β-glucan.

The abstract seems to overpromise a bit. First, it mentions trained immunity and HSCs, but they don't seem to formally address either in the context of this model (there is reprogramming as assessed by transcriptome and metabolic analyses which is suggestive as stated by the authors, but do the changes overlap significantly with classically trained immunity?), and second, it links phenotypes together in a pathway(s) that they haven't actually interrogated - although they look at transcripts and do a seahorse assay they don't actually confirm that any of those findings are related to the increased response to LPS in vivo. The long discussion with all the caveats highlights these limitations, all relegated to future studies.

We thank the reviewer for this comment. In response, we have revised the abstract to more accurately highlight the key findings of this study. Specifically, we introduced the concept of central trained immunity to describe the phenomena commonly observed with β-glucan treatment, contrasting it with the peripheral trained immunity detailed in the manuscript.

The use of Csf2rb-/- mice to complement the clodronate approach is interesting (this approach has been used in the past with influenza virus). In addition to lacking AMs, these mice develop pulmonary alveolar proteinosis. Do the authors have histopathology from these mice in the current model? They mention PAP in the discussion.

Pulmonary alveolar proteinosis (PAP) typically develops in Csf2b-/- mice from 12 weeks of age onwards (Stanley et al., Proc Natl Acad Sci USA, 1994). However, in our model, mice were euthanized at 6 weeks, ensuring that pulmonary function and structure remained intact. A hallmark of PAP is the accumulation of protein, primarily surfactant, in BAL. To investigate this, we measured BAL protein concentration and observed no differences at baseline (Figure 2F). These findings were further supported by the absence of differences in BAL proinflammatory cytokine concentrations (Figure 2H).

A question about their BAL technique? In the control mice without glucan/LPS stimulation, only 40% of BAL cells are AMs and the total number of AMs (range of <103 to 2-3 x 104) is at least 5-fold lower than typically seen in BALs from healthy mice (105), and there didn't seem to be many PMNs either. Are 60% of the BAL cells lymphocytes/ RBCs? Is it possible that overall AM numbers are changing, but CD11c/SiglecF-positive cell numbers stay the same (only assessed 2 markers)? More phenotyping would help.

We appreciate the reviewer’s comment and would like to clarify that alveolar macrophages (AMs) are presented in the manuscript as a frequency of viable cells rather than as a frequency of CD45^+^ cells, to ensure consistency throughout the study. The remaining cells in the samples are likely epithelial cells and lymphocytes, as red blood cells are lysed during sample processing. For additional context, we now provide data showing AMs as a percentage of CD45^+^ cells, which account for 80–90% of leukocytes. Furthermore, in New Supplementary Figure 1D, we highlight the expression of AM-associated markers to better define their phenotype. We observed no differences in MHCII, CD169, CD64, or F4/80 expression in βglucan-treated mice. However, there was an increase in CD80^+^ AMs, indicating enhanced activation and corroborating their proinflammatory phenotype.

**Author response image 1. sa3fig1:** AMs as percentage of CD45^+^ cells. Mice were treated with β-glucan for seven days. We show CD11c^+^SiglecF^+^ cells in the bronchoalveolar lavage (BAL) as a percentage of CD45^+^ cells (n=5).

Line 130-131. TNF is decreased and not pointed out.

In the poly(I:C) model, the difference in the BAL TNF concentration is not statistically different between naïve and trained mice due to high variability of data. The reviewer is correct that TNFα does not appear to reflect Poly(I:C)-mediated ALI. We have included this point in the revised manuscript (Line 146-148).

**Reviewer #2 (Recommendations for the authors):**
Suggestions:(1) The authors provide evidence for enhanced ALI via different techniques, e.g. histology, vascular leakage, immune cell composition in BAL etc. It would be interesting to see whether there were any changes in the disease severity of ALI. If possible the authors could provide data for survival, temperature, weight, and/or glucose in the different groups.

Mice are extremely resistant to the pulmonary LPS model. We have previously assessed lethality of our LPS model, and all mice survive even with an increased intranasal dose of LPS 200μg (Pernet et al, Nature, 2023). To address the reviewer concerns, we next assessed the morbidity by monitoring weight loss following LPS challenge and showed β-glucan-treated mice exhibit a delayed recovery time after 4 days LPS treatment (New Supplementary Figure 1B).

(2) The authors show that ß-glucan mediated training enhances ALI. Conversely, the opposite, decreased immunopathology should be observed in case an LPS tolerance model would be used. I am wondering whether this has already been performed, given that the (LPS/immune)tolerance field is already older than the training field. If not, I suggest incorporating this feature in their discussion.

Thank you for this insightful comment. While LPS has long been recognized to induce tolerance, studies have also shown that intranasal exposure to ambient levels of LPS can induce alveolar macrophage (AM) training via type I interferon signaling (Zahalka et al., Mucosal Immunol, 2022). In contrast, Mason et al. demonstrated that systemic LPS stimulation induces tolerance through TNF-α signaling, resulting in diminished AM phagocytosis and superoxide production. This leads to reduced neutrophil recruitment and impaired bacterial clearance in a *Pseudomonas aeruginosa* pneumonia model (J Infect Dis, 1997). Furthermore, we recently reported that systemic administration of β-glucan induces central trained immunity, generating a distinct subset of regulatory neutrophils that promote disease tolerance against influenza viral infection (Khan et al., Nat Immunol, 2025). These findings highlight the complex and context-dependent interplay between training and tolerance. We have expanded on this point in the discussion section of the revised manuscript (Lines 289-297).

(3) The finding that trained immunity can exert not only beneficial effects but also enhance immunopathology is interesting and should be further explored. Already Jentho et al. (PNAS 2021) have shown that upon sterile inflammation as imposed by LPS, (heme) training can lead to enhanced mortality. This might be a relevant trade-off in trained immunity since no beneficial resistance effect by pathogen killing can be obtained. It would be interesting to see, in their model, whether heme would also enhance ALI after intranasal LPS application. Or at least, can the authors discuss this finding more, also in relation to the already published evidence?

Thank you for raising this interesting point, which is indeed relevant to our study. Jentho et al. demonstrated that training by heme can be beneficial in combating infectious challenges but can have deleterious effects in the context of sterile inflammation. The concept of endogenous training agents like heme, with their diverse effects on immune cells, aligns well with our βglucan model, particularly given the high prevalence of fungal agents in the microbiome.

While investigating the effects of heme on alveolar macrophages would certainly be intriguing, Jentho and colleagues have already reported the maladaptive effects of heme, such as tissue damage, during sterile LPS-induced inflammation. As such, these findings might be redundant in the context of our model. However, we have drawn a relevant parallel and expanded on this discussion in the revised manuscript (Lines 382-385).

(4) It is not clear how the histologies were evaluated. This is a field of great subjectivity. The authors should describe it in more detail. The best option would have been a blinded observer. Was this done?

Histology samples were evaluated according to ATS 2011 guidelines regarding “Features and measurements of experimental acute lung injury in animals” by a blinded pathologist. We have specified this in the methods of the revised manuscript.

Minor:(1) Line 108 and ff. Please change TNF, not TNFa

Since we used an ELISA specific for TNF-α rather than general TNF, it is more accurate to refer to it as TNF-α.

(2) Line 513 and ff. Please use Greek letters when appropriate, e.g. IFN-γ not IFNg.

Thank you for pointing out these mistakes, we rectified these in the text.